# Electrically empowered microcomb laser

Jingwei Ling [1,3], Zhengdong Gao[1,3], Shixin Xue[1], Qili Hu [2], Mingxiao Li [1], Kaibo Zhang[2], Usman A. Javid [2], Raymond Lopez-Rios[2], Jeremy Staffa [2] & Qiang Lin [1,2] ✉

Optical microcomb underpins a wide range of applications from communication, metrology, to sensing. Although extensively explored in recent years, challenges remain in key aspects of microcomb such as complex soliton initialization, low power efficiency, and limited comb reconfigurability. Here we present an on-chip microcomb laser to address these key challenges. Realized with integration between III and V gain chip and a thin-film lithium niobate (TFLN) photonic integrated circuit (PIC), the laser directly emits mode-locked microcomb on demand with robust turnkey operation inherently built in, with individual comb linewidth down to 600 Hz, whole-comb frequency tuning rate exceeding $2.4 \times 10^{17}$ Hz/s, and 100% utilization of optical power fully contributing to comb generation. The demonstrated approach unifies architecture and operation simplicity, electro-optic reconfigurability, high-speed tunability, and multifunctional capability enabled by TFLN PIC, opening up a great avenue towards on-demand generation of mode-locked microcomb that is of great potential for broad applications.

Optical frequency comb is a coherent light source that consists of many highly coherent single-frequency laser lines equally spaced in the frequency domain. Its development has revolutionized many fields including metrology, spectroscopy, and clock[1]. In recent years, significant interest has been attracted to the generation of phase-locked optical frequency comb in on-chip nonlinear microresonators[2–4]. The superior coherence offered by these mode-locked microcombs has rendered a variety of important applications including data communication[5], spectroscopic sensing[6], optical computing[7,8], range measurement[9–12], optical[13] and microwave[14] frequency synthesis, with many others expected in the years to come.

Despite this great progress, challenges remain in the development and application of microcombs. The first is the difficulty in triggering comb mode-locking due to the intrinsic device nonlinearities. Recently, self-starting operations have been demonstrated to address this issue[15–18]. Their implementations, however, require sophisticated system pre-configuration and careful balance of specific nonlinear dynamics, which are difficult to apply in most practical devices. The second is the low power efficiency of soliton microcomb generation due to the pump-laser-cavity frequency detuning induced by soliton pulsing. Although pulse pumping[19] or auxiliary-resonator enhancement[20,21] can improve

the generation efficiency, they require delicate synchronization in time or resonance frequency and the difficulty of soliton initialization remains the same. The third is the limitation in the comb controllability due to the monolithic nature of the comb generator that is difficult to change after the device is fabricated. Piezoelectric effect could be used to deform the comb resonator[12], which, however, exhibits limited tuning speed and efficiency due to its slow mechanical response. To date, the majority of comb generators still have to rely on external laser control to adjust the microcomb state.

Recently, there are significant advances in chip-scale integration of semiconductor lasers and nonlinear comb generators[16,22–24], in which a diode laser produces single-frequency laser emission to pump a hybridly or heterogeneously integrated external nonlinear resonator to excite microcombs. Such a fully integrated system shows great promise in improving the size, weight, and power consumption. However, the nature of soliton comb generation remains essentially the same, with all the above challenges persistent. Up to now, the realization of an integrated comb source free from these challenges remains elusive.

Here we present a fundamentally distinctive approach to resolve these challenges in a single device. Figure 1a shows the device concept.

[1]Department of Electrical and Computer Engineering, University of Rochester, Rochester, NY, USA. [2]Institute of Optics, University of Rochester, Rochester, NY, USA. [3]These authors contributed equally: Jingwei Ling, Zhengdong Gao. ✉e-mail: qiang.lin@rochester.edu

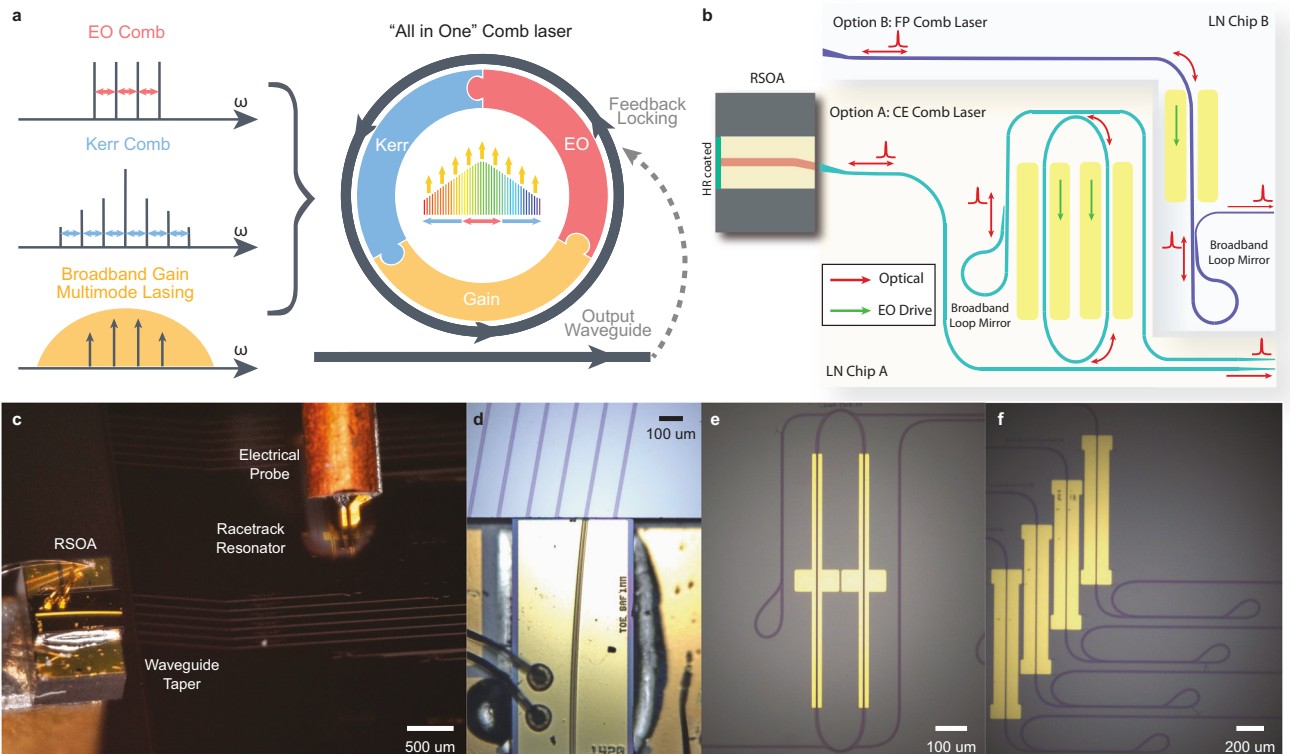

**Fig. 1 | Device concept of the integrated comb laser. a** Conceptual illustration of the comb generation and mode-locking principle, in which electro-optic (EO) comb generation, Kerr comb generation, and broadband optical gain all work synergistically together inside a single laser cavity for on-demand generation of mode-locked soliton comb. In addition, the laser comb output is detected and fed back to the laser cavity for resonant EO modulation to realize a self-sustained operation. **b** Schematic of comb laser cavity structure formed by hybrid integration between a RSOA chip and a LN external cavity chip. Two different configurations are employed: A, cavity-enhanced (CE) comb laser structure in which the LN external cavity is formed mainly by an embedded high-Q racetrack resonator together with a broadband Sagnac loop mirror; B, Fabry-Perot (FP) comb laser in which the LN external cavity is formed by an EO phase modulation section together with an a broadband Sagnac loop mirror. **c** Photo of a CE comb laser, showing that the RSOA is edge-coupled to the LN external cavity chip. **d** Zoom-in photo showing the edge-coupling region between the RSOA and the LN chip. **e** Photo of the racetrack resonator and the loop mirror in a CE comb laser. **f** Photo of the EO phase modulator and the loop mirror in an FP comb laser.

In contrast to conventional approaches that rely solely on a single mechanism—either optical Kerr or electro-optic effect—for comb generation while with external pumping, we utilize the resonantly enhanced electro-optic (EO) modulation to initiate the comb generation, the resonantly enhanced optical Kerr effect to expand the comb bandwidth and phase-lock the comb lines, and the embedded III-V optical gain to sustain and stabilize the comb operation. Moreover, the resulting coherent microwave (via optical detection) is fed back to the EO comb to further enhance the mode-locking, leading to unique self-sustained comb operation.

We realize this approach by integrating a III-V gain element with a thin-film lithium-niobate (LN) photonic integrated circuit (PIC) to produce a III-V/LN comb laser (Fig. 1b). LN PIC has attracted significant interest recently[25–28] for a variety of applications including high-speed modulation[29,30], frequency conversion[31–33], optical frequency comb[15,34,35], and single-frequency lasers[36–39]. Here, we unite active EO modulation with passive four-wave mixing (FWM) in a dispersion-engineered high-Q laser cavity for the on-demand generation of mode-locked soliton microcomb, which naturally leads to self-starting full turnkey operation simply by turning on/off either the RF signal driving the comb resonator or the electric current driving the gain element. As the comb modes extract energy directly from material gain, all of the optical power obtained from the III-V gain medium contributes fully to the comb generation, distinct from conventional microcombs in which the majority of the optical power remains in the pump wave. Moreover, the strong electro-optic effect of the LN cavity enables high-speed tunability

of comb frequencies and electro-optic reconfigurability of comb spectrum and mode spacing. With this approach, we are able to produce broadband highly coherent microcombs, with individual comb linewidth down to 600 Hz, frequency tunability of over $2.4 \times 10^{17}$ Hz/s for the entire microcomb, microwave phase noise down to -115 dBc/Hz at 500 kHz frequency offset, and a wall-plug efficiency exceeding 5.6%. The simplicity of the demonstrated approach opens up a new path for on-demand generation of mode-locked microcombs that is expected to have profound impact on the broad applications in high-precision metrology, telecommunications, remote sensing, clocking, computing, and beyond.

## Results

### Comb laser structure design

The III-V/LN microcomb laser is formed by integrating an InP reflective semiconductor optical amplifier (RSOA) with an LN external cavity chip via facet-to-facet coupling. We employ two types of laser cavity structures for the purpose, as shown in Fig. 1b. Chip A laser structure is embedded with a dispersion-engineered EO microresonator and Chip B laser structure consists of a simple EO phase modulation waveguide section. The advantage of Chip-A resonator-type structure is that the high-Q microresonator offers strong cavity enhancement for comb generation and mode-locking, which we term as the cavity-enhanced (CE) comb laser. The benefit of the Chip-B Fabry-Perot-type structure is that it offers flexibility of mode-locking operation, which we term as the Fabry-Perot (FP) comb laser.

In the CE comb laser, the group-velocity dispersion (GVD) of the racetrack microresonator (intrinsic optical Q ~ $1.6 \times 10^6$) is engineered to be small but slightly anomalous to support broadband comb generation. At the same time, the pulley coupling regions are specially designed for uniform close-to-critical coupling to the resonator over a broad telecom band. Such a design ensures high loaded optical Q ($\sim 5 \times 10^5$) of the resonator uniformly across a wide spectral range which is crucial both for enhancing the comb generation and mode-locking and for efficient light coupling into/out of the microresonator. Moreover, we also engineer the GVD of straight waveguide sections outside the racetrack resonator to compensate for that of the RSOA section so as to minimize the overall laser cavity GVD. At the same time, the overall optical path length of the laser cavity is designed to be an integer multiple of the racetrack resonator's for matching their resonance mode frequencies and round-trip group delay. The GVD of the FP comb laser is engineered in a similar fashion. The free-spectral range (FSR) of the racetrack resonator is designed to be around 40 GHz to better accommodate bandwidths of RF filter and amplifier after detection of comb beating signal. In contrast, the FP comb laser is designed to have a small FSR of around 10 GHz for easy operation of harmonic mode locking.

To support the broadband operation, the Sagnac loop mirror employs an adiabatic coupling design[40] to achieve high reflection and feedback with a reflectivity of > 95% over a broad spectral band of 1500–1600 nm. On the other hand, a horn taper waveguide is designed on the LN chip to minimize the coupling loss between the LN chip and the RSOA gain chip. The RSOA exhibits a broadband gain in the telecom L-band, with a 3-dB bandwidth of > 40 nm. Figure 1c–f shows the device structures. Details about the design parameters and the characterization of the laser structures are provided in the Supplementary Information (SI).

## Comb laser performance

To excite the mode-locked microcomb, we first launch a single-frequency RF signal to drive the racetrack resonator of the CE comb laser (Fig. 2a), with a frequency of 39.58 GHz that matches its FSR. Before the RF signal is applied, the device exhibits single-mode or multi-mode lasing, with an example shown in the blue curve of Fig. 2b. However, A microcomb is readily produced as soon as the RF signal is applied, with an optical spectral bandwidth of about 20 nm (Fig. 2b, red curve). Mode-locking of the comb is verified by the clean RF tone at 39.58 GHz detected from the beating between comb lines (Fig. 2c), as well as the autocorrelation trace from the laser output pulses (Fig. 2b, inset). The 39.58-GHz RF tone exhibits a high signal-to-noise ratio of 79 dB (Fig. 2c, inset), whose phase noise spectrum matches identically the driving RF source (Fig. 2d), showing the preservation of the relative phase coherence between comb lines via mode-locking. Mode-locking of the comb is also clearly evident by the clean noise floor around the DC region in the RF spectrum (Fig. 2c see SI for details), where the zero extra noise from the mode-locked comb state infers that all the comb lines of the entire comb are phase-locked together.

The underlying mechanism responsible for mode-locking dominantly contributes from the combined resonant EO modulation and optical Kerr effect, in which the EO modulation produces EO sidebands to initiate the comb generation while the optical Kerr effect broadens the comb spectrum and phase-locks the comb lines (Fig. 1a). This unique comb generation mechanism distinguishes the comb laser from other approaches[15–21] (see SI for more details). Indeed, the laser is able to produce mode-locked soliton pulses in the absence of EO modulation (while with a narrower spectrum), in which only the optical Kerr effect is responsible for mode-locking. The detailed theoretical modeling and testing results are provided in SI. In Fig. 2c, the small RF tone around the half-harmonic at 19.79 GHz indicates certain comb dynamics. It can be eliminated by reconfiguring the laser and one example is shown in SI which exhibits a clean single RF beating tone

and a well-defined sech²-shaped soliton pulse spectrum. The two lasers mainly differ in their overall dispersion of the laser cavity, indicating that the device dispersion plays an important role on the comb spectrum. The two-sidelobe feature of the comb spectrum in Fig. 2b implies that the output pulses are likely to be mode-locked two-color pulses in which the two color pulses bounds with each other via certain inter-pulse interaction[41]. Its exact nature, however, will require further exploration. The details for comb spectrum reconfiguration based RF tuning are provided in the SI.

In addition to the high coherence between the comb lines, the comb laser also exhibits narrow linewidth on its individual comb lines. To show this feature, we employ the correlated self-heterodyne method[42,43] to characterize the overall linewidth of the whole comb laser by launching the entire comb for linewidth measurement (rather than characterizing individual comb lines themselves) (See SI for details). The recorded phase noise spectrum is shown in Fig. 2e, which indicates a white frequency-noise floor of ~ 350 Hz²/Hz (Fig. 2e, inset) that corresponds to a laser linewidth of ~ 2 kHz. The linewidth of the comb lines can be decreased further and an example is shown in Fig. 2e for a slightly different comb state produced from the same laser, which exhibits a white frequency-noise floor of ~ 100 Hz²/Hz (Fig. 2e, inset) that corresponds to a laser linewidth as low as ~ 600 Hz. Note that these values represent the overall linewidth contributed from the entire comb, which indicates the averaged intrinsic linewidth of individual comb line.

Figure 2f shows the current-dependent characteristics of the comb laser, which exhibits a low threshold current of 50 mA, indicating the low overall loss of the integrated laser. The comb laser produces an optical output power of 11 mW, in which the major individual comb lines exhibit a power in the range of 0.25–2 mW. The power is measured at a pumping current of 275 mA and a pumping voltage of 1.4 V, which corresponds to a wall-plug efficiency of 2.8%, defined as the ratio between the output optical power and the electrical power used to drive the RSOA. As the laser has two output ports (Fig. 1b) that emit the same amount of optical power, the total wall-plug efficiency of the laser is thus 5.6%. This level of wall-plug efficiency is on par with other integrated external-cavity semiconductor lasers recently developed[44,45]. Intriguingly, the comb power increases with increased driving RF power, whose details are provided in the SI. Note that the total optical power contributes fully to the generated comb, in strong contrast to conventional Kerr solitons or EO combs in which the major optical power remains in the residual pump wave with low comb generation efficiency.

A distinctive feature of the comb laser is that the produced comb can be switched on/off on demand by simply switching on/off the driving RF signal. To show this feature, we beat the comb with a reference single-frequency laser operating at the wavelength of 1582 nm that is inside the comb spectrum, and monitor the beating signal with the RF driving signal being turned on/off. As shown in Fig. 2g and i, the beating signal follows faithfully the driving RF signal. The coherent beating signal shows up readily when the RF driving is on, indicating the generation of the mode-locked comb. The beating signal disappears right after the RF driving is off, indicating the shutoff of the comb state. Same phenomenon is observed when the reference laser is tuned to other wavelengths within the comb spectrum.

Similar phenomena are observed in the FP comb laser, while generally with smaller spectral extents due to the lack of cavity enhancement. The FP comb laser, however, exhibits a distinctive feature in that it can be flexibly mode-locked at higher harmonics of the laser cavity FSR. Figure 3 shows this feature. We are able to achieve third- and fourth-order harmonic mode-locking by applying an RF signal to the phase modulation section of the FP comb laser, with a frequency of 29.45 and 39.27 GHz, respectively, that are three and four times of the laser FSR (9.817 GHz). Again, mode locking of the combs is clearly verified by the detected RF beating signal from the combs with

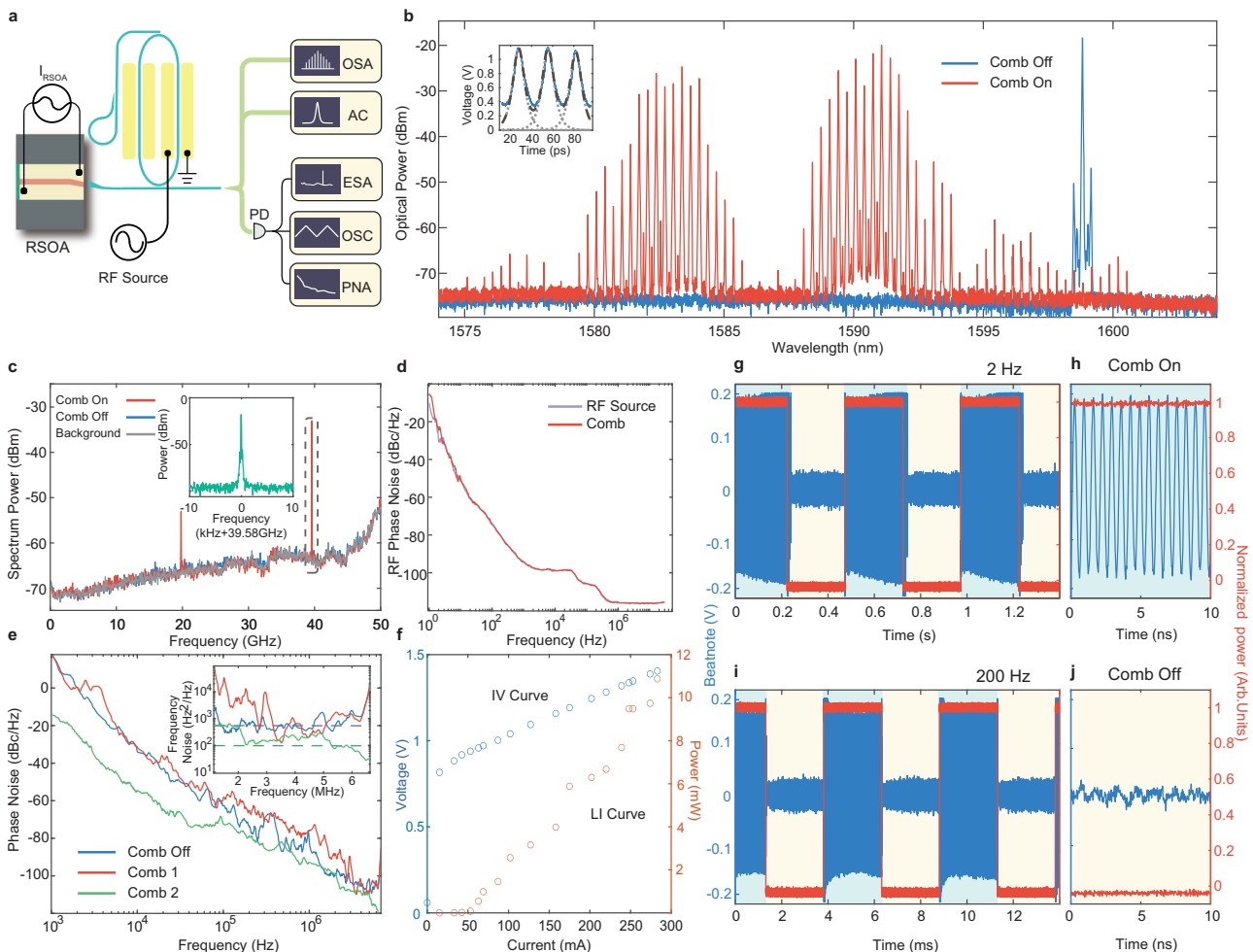

**Fig. 2 | Lasing performance of a CE comb laser. a** Schematic of the experimental setup for comb laser characterization. OSA: optical spectrum analyzer; AC: autocorrelator; PD: photo-detector; ESA: electrical spectrum analyzer; OSC: real-time oscilloscope; PNA: phase noise analyzer. **b** Optical spectrum of the comb laser output. Off state: single mode lasing with the RF driving off. On state: comb lasing with the RF driving on. Inset: Autocorrelation trace of the laser output pulses, in the "comb on" state, in which the blue curve shows the experimental data, dotted curves show the the fitted autocorrelation profiles from individual sech² pulses, and the dashed curve show the overall fitted autocorrelation trace. The autocorrelation is recorded directly from the laser output pulses without dispersion compensation. **c** Electrical spectrum of the beat note detected from the comb laser output. The red and blue curves show the comb-on and comb-off (single-mode lasing) states, respectively, corresponding to those in **b**. Gray curve shows the noise background

of the optical detector, as a comparison. The inset shows the detailed spectrum of the RF beat note at 39.58 GHz. **d** Phase noise spectrum of the 39.58-GHz RF beat note (red) and the RF driving signal (blue). **e** Phase noise spectrum of the CE comb laser output measured with a self-heterodyne method. Red and green curves show for two different comb states, and blue curve shows for the comb-off (single-mode lasing) state. Inset shows the corresponding frequency noise spectrum of the three laser states. **f** P–I–V curve of the CE comb laser, in which the red and blue curves show the L-I and I-V curves, respectively. **g–j** Turnkey operation of the comb laser at two different speed of 2 Hz **g**, **h** and 200 Hz **i**, **j**, respectively. Red curve shows the normalized driving RF power and blue curve shows the beating signal between the comb laser ouput and an external reference laser at 1582 nm. **h**, **j** show the zoomed-in signal for the on/off states, respectively.

a SNR of 77 dB, as well as by the autocorrelation traces from the laser output pulses (Fig. 3b,c, insets).

## High-speed frequency tuning of the comb laser

Another distinctive characteristic of the comb laser is that the laser frequencies of the entire mode-locked comb can be tuned cohesively at a high speed. To show this feature, we apply a triangular-waveform electric signal—together with the 39.58-GHz RF driving signal—to the racetrack resonator of the CE comb laser as shown in Fig. 4a. While the 39.58-GHz RF driving signal supports the mode-locking process, the triangular-waveform electric signal will adiabatically tune the resonance frequencies of the racetrack resonator, thus tuning the laser frequencies of the entire mode-locked comb together as a whole.

To show this feature, we beat the comb with a narrow-linewidth reference CW laser at 1582 nm that is about 15 GHz away from a comb line, and monitor the beating signal in real time. At the same time, we

monitor the spectrum of the recorded 39.58-GHz RF tone from the beating between the comb lines (See SI for details of the setup). The frequency dynamics of the 15-GHz beating signal with the reference laser show the frequency tuning of the comb line nearby while the 39.58-GHz RF tone from the comb line beating indicates the quality of mode locking during the frequency tuning. Figure 4b–d shows the temporal variation of the 15-GHz beating signal at different modulation speeds of 1, 10, 100 MHz. They show clearly that the frequency tuning of the comb line follows faithfully the waveform of the driving triangular-waveform electric signal at all modulation speeds, with a deviation of no more than 5%. In particular, the recorded 39.58-GHz RF tone from the comb line beating (Fig. 4e–g) remains unchanged during the frequency tuning, except with created modulation sidebands that simply results from the laser frequency modulation (see also Fig. 4a, right figure). This observation confirms that the phase-locking between the comb modes is fully preserved during the high-speed

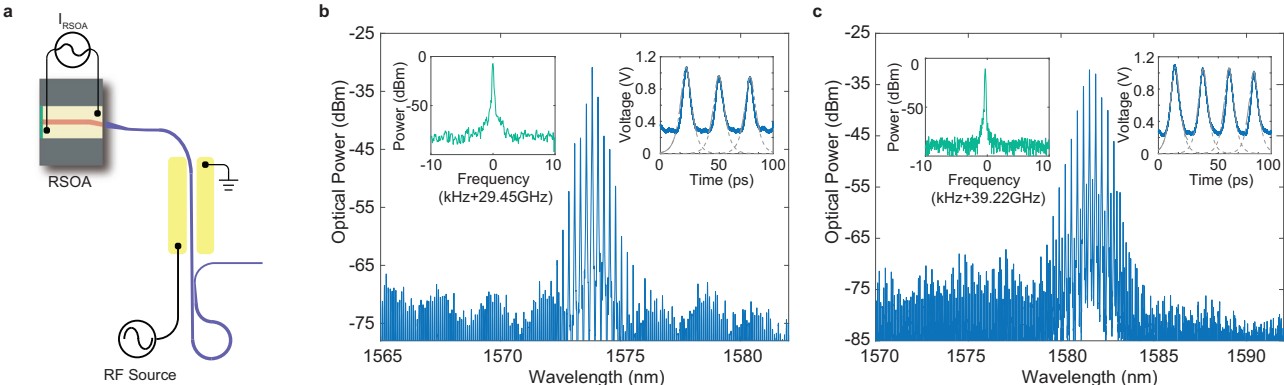

**Fig. 3 | Harmonic mode-locking of a FP comb laser with an FSR of 9.817 GHz.** Third harmonics (29.45 GHz) and fourth harmonics (39.27 GHz) are separately used as driving signals. **a** Schematic of the experimental setup for harmonic mode-locking of the comb laser. **b** Optical spectrum of the laser output with third-harmonic mode-locking, by driving the phase modulator with a RF signal at 29.45 GHz. **c** Optical spectrum of the laser output with fourth-harmonic mode-locking, by driving the phase modulator with a RF signal at 39.27 GHz. In **b**, **c**, the left insets shows the electrical spectrum of the RF beat note detected from the output laser comb, and the right insets show the autocorrelation trace of the laser output pulses with dashed curve showing the fitted individual pulses. Same as Fig. 2b, auto-correlation is recorded directly from the laser output pulses without dispersion compensation or pulse shaping.

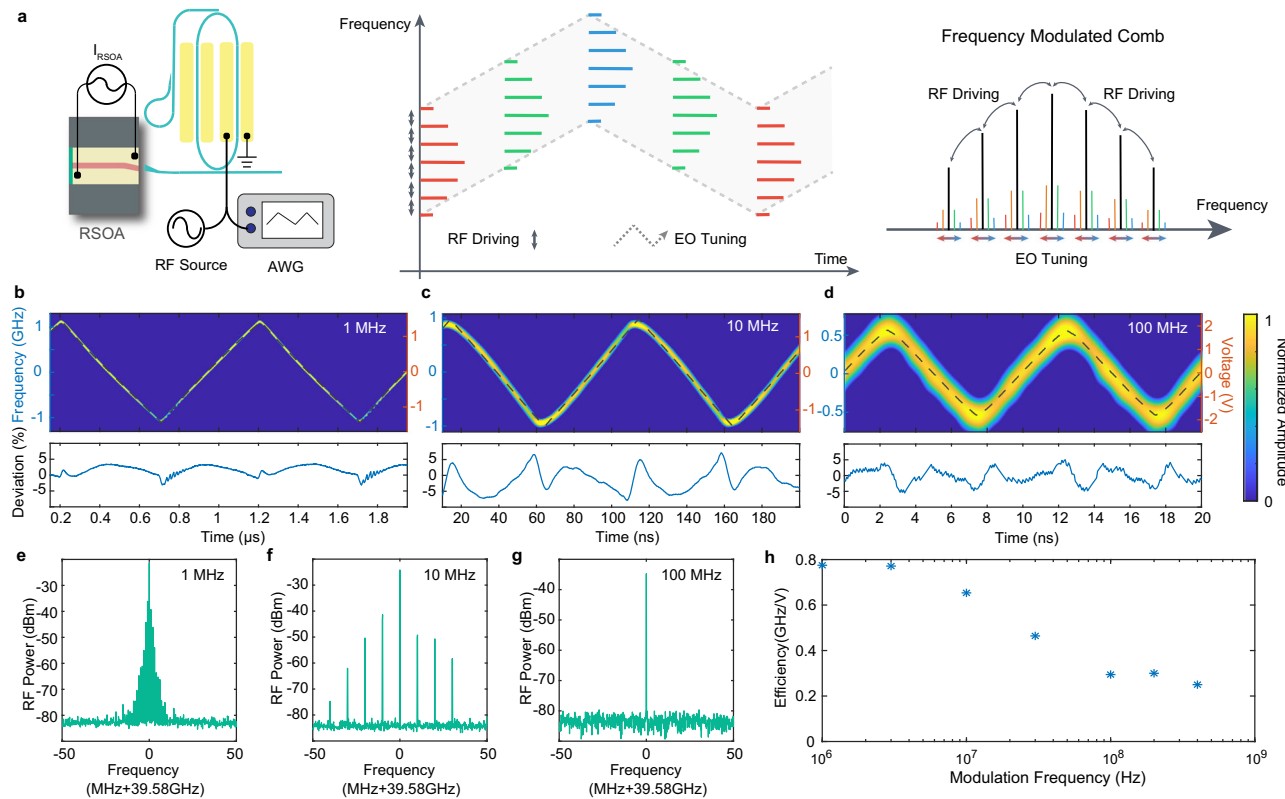

**Fig. 4 | Fast frequency tuning of the whole lasing comb. a** Left panel: Schematic of the setup for comb frequency tuning, in which a triangular-waveform electrical signal produced by an arbitrary waveform generator (AWG) is used to drive the racetrack resonator of the CE comb laser together with the 39.58-GHz mode-locking RF signal. Middle panel: Conceptual illustration of the comb frequency tuning process, showing the laser frequencies of the comb are tuned together as a whole. Right panel: Schematic showing the corresponding sideband creation around the comb lines, introduced by triangular-waveform frequency modulation. **b**–**d** Time-frequency spectra of the beatnote between the comb laser output and a referenced laser operating at a fixed wavelength of 1582 nm, at the modulation speed of 1, 10, and 100 MHz, respectively. The dashed curves show the corresponding triangular-waveform EO tuning signal. Bottom panels: Corresponding relative frequency deviation at each modulation speed. **e**, **f**, **g**. Electrical spectrum of the 39.58-GHz beat note detected from the laser output comb, at modulation speed of 1 MHz **e**, 10 MHz **f**, and 100 MHz **g**, respectively. **h**. Laser frequency tuning efficiency recorded at different modulation speeds.

frequency tuning process, indicating that the entire mode-locked comb is tuned in its frequencies as a whole, without any perturbation to the comb mode spacing. This is in strong contrast to other comb modulation approaches[11,12,46] where the comb mode spacing is seriously impacted by external modulation. The frequency tuning range of 1.2 GHz at the modulation speed of 100 MHz (Fig. 4d) corresponds

to a frequency tuning rate as high as $2.4 \times 10^{17}$ Hz/s for the comb. Both the frequency tuning rate and tuning speed are orders of magnitudes higher than the piezoelectric tuning and the external pump modulation approaches[11,12], which are constrained only by the photon lifetime of the high-Q racetrack resonator. As shown in Fig. 4h, the device exhibits a frequency tuning efficiency of about 0.2–0.8 GHz/V

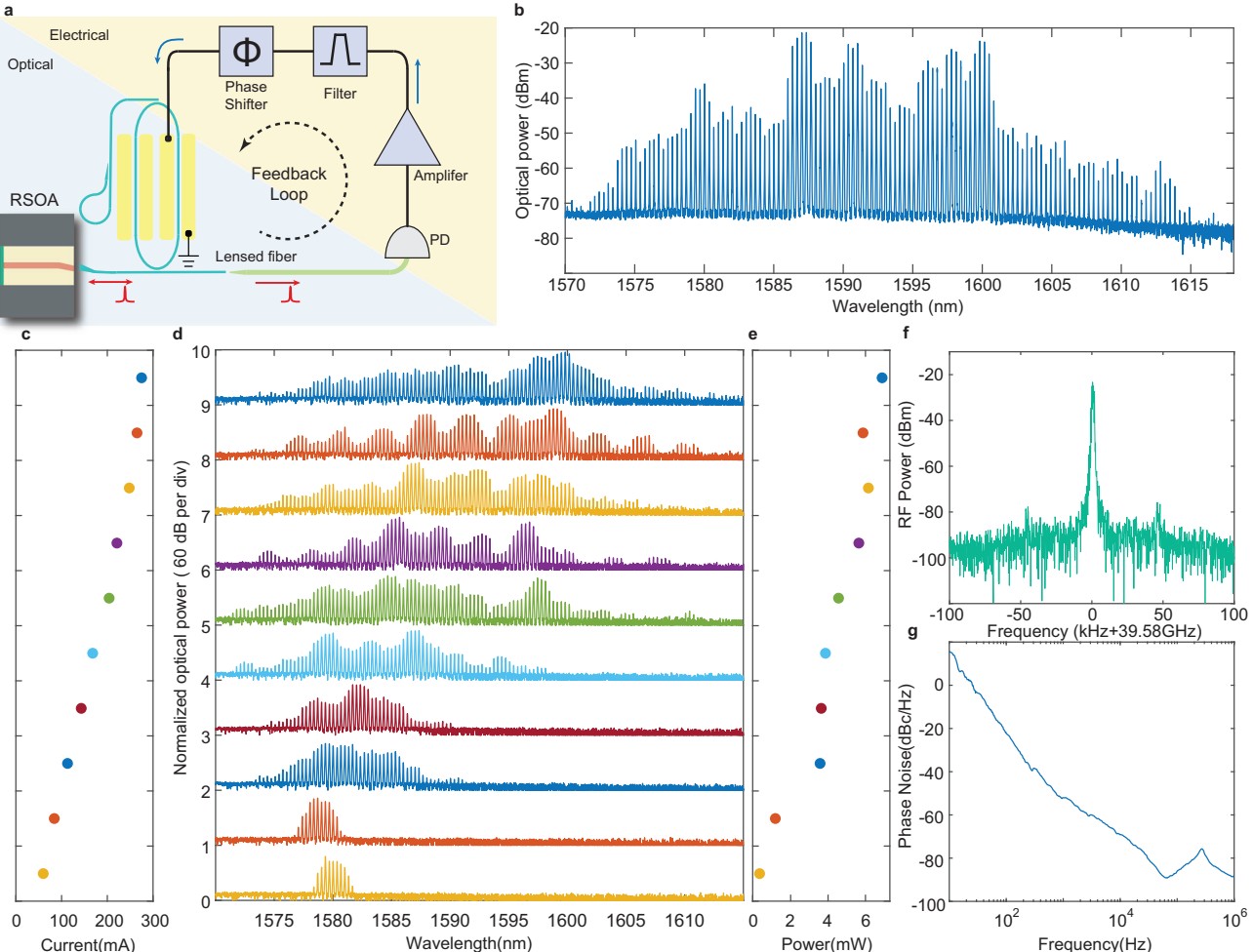

**Fig. 5 | Self-sustained operation of the CE comb laser with feedback locking.**
**a** Schematic of self-feedback locking of the CE comb laser. **b** Optical spectrum of the comb laser output at a driving current of 285 mA. **c**–**e** Optical spectrum **d** and optical power **e** of the comb laser output as a function of the RSOA driving current **c. f** Electrical spectrum of the 39.58-GHz beat note detected from the laser output comb, with a driving current of 60 mA. **g** Phase noise spectrum of the detected 39.58-GHz beat note.

depending on the modulation speed, which is more than an order of magnitude higher than the piezoelectric approach[12]. The tuning efficiency can be further doubled by employing both sets of driving electrodes of the racetrack resonator (Fig. 4a).

### Feedback mode-locking of the comb laser
So far, the comb laser utilizes an external RF signal to support the mode-locking. This signal, however, can be removed by feeding the coherent 39.58-GHz RF tone detected from the comb mode beating directly back to the comb laser cavity to sustain the mode-locking, resulting in unique stand-alone self-sustained comb lasing operation. Figure 5a illustrates this approach. The comb laser output is detected by a high-speed optical detector whose output RF signal is amplified to an adequate amplitude, filtered to suppress excess low-frequency noises, adjusted with appropriate phase, and then fed back to drive the racetrack resonator of the CE comb laser. This approach is somewhat similar to[47,48] but the feedback modulation here is directly applied to the enhancing microresonator inside the laser cavity. The combined EO and optical Kerr effects lead to significantly broader comb spectrum, in contrast to the pure EO comb in[47,48].

As shown in Fig. 5b, a broadband microcomb with a spectrum covering about 50 nm and an optical power of 8.5 mW is produced on chip with a driving current of 285 mA. Indeed, the microcomb is readily produced on demand as soon as the driving current is turned on, with a driving current as low as 60 mA, as shown in Fig. 5c–e. Mode locking of the comb is clearly evident by the clean 39.58-GHz RF tone detected from the comb mode beating (Fig. 5f), which exhibits a SNR of 65 dB and a narrow 3-dB linewidth of 1.5 kHz. The phase noise of the RF beating signal reaches a level of -90 dBc/Hz at an offset frequency of 60 kHz, which is considerably lower than the laser heterodyne beating approach[49] and is comparable to that of free-running optical Kerr soliton microcombs[46,50,51]. The optical spectral bandwidth and the output power of the comb laser increase considerably with increased driving current (Fig. 5d). This is expected since the increased optical power of the mode-locked comb inside the high-Q racetrack resonator would significantly enhance the optical Kerr effect and the resulting four-wave-mixing process to broaden the comb spectrum. No saturation is observed on the comb spectral bandwidth as the current increases.

## Discussion
The attainable extent of the microcomb spectrum or soliton pulse width in current devices is primarily limited by the available optical power inside the cavity and the group delay mismatch between the enhancing resonator and the main laser cavity. For the former, it can be improved by either reducing the loss (*e.g.*, improving the RSOA-LN chip coupling efficiency) or increasing the optical gain (*e.g.*, using a higher-power RSOA) inside the laser cavity. For the latter, our

theoretical modeling (see SI) shows that the formation of ideal ultrashort soliton pulses would require that the roundtrip time of the main laser cavity be integer times that of the enhancing racetrack resonator. In current devices, however, there is a certain amount of mismatch which limits the comb spectrum and the coherence of the mode-beating RF tone. This problem can be resolved by further optimization of the roundtrip length of the main laser cavity, for example, via heterogeneous integration[24] in which the roundtrip length of the main laser cavity can be precisely defined by the fabrication process. It can also be resolved by introducing a certain group-delay tuning element[52] into the external laser cavity. With these optimizations, we expect that ultra-broadband highly coherent soliton microcomb can be produced.

To conclude, we have introduced a chip-scale microcomb laser that can be flexibly mode-locked with either active-driving or passive-feedback approaches and that can be tuned/reconfigured at an ultra-fast speed, with robust turnkey operation inherently built in. The demonstrated integrated comb laser exhibits remarkable reconfigurability and performance significantly beyond the reach of conventional on-chip mode-locked semiconductor lasers[53–56]. The demonstrated devices combine elegantly the simplicity of integrated laser structure, robustness of mode-locking operation, and electro-optically enhanced tunability and controllability, opening up a new avenue towards on-demand generation of soliton microcombs with high power efficiency that we envision to be of great promise for a wide range of applications including ranging, communication, optical and microwave synthesis, sensing, metrology, among many others.

## Methods

### Device fabrication

The device fabrication begins with a congruent x-cut thin film lithium-niobate-on-insulator (LNOI) wafer, with a 600 nm LN layer on a $4.7\,\mu m$ silica-coated silicon substrate. E-beam lithography (EBL) and Ar-ion milling are used to etch the waveguide with ZEP-520A as mask. Etching thickness ranges from 300 nm (CE comb laser) to 350 nm (FP comb laser) for dispersion engineering. Second EBL is applied on PMMA for deposition of 400 nm gold-evaporated electrodes, which are placed $2.5\,\mu m$ from the waveguide. The distance between the waveguide and electrode is chosen to balance the EO modulation frequency with loss from metal absorption. Dicing and polishing of LN chip are employed at last to acquire optimized fiber-to-chip and amplifier-to-chip coupling, with both coupling losses around 6 dB.

## Data availability

All data are available in the main text or the supplementary information.

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

## Acknowledgements

We would like to thank Lin Chang and William Renninger for their helpful discussions. This work is supported in part by the Defense Advanced Research Projects Agency (DARPA) QuICC program under Agreement No. FA8650-23-C-7312 and LUMOS program under Agreement No. HR001-20-2-0044, and the National Science Foundation (NSF) under Grant No. OMA-2138174 and ECCS-2231036. This work was performed in part at the Cornell NanoScale Facility, a member of the National Nanotechnology Coordinated Infrastructure (National Science Foundation, ECCS-1542081); and at the Cornell Center for Materials Research (National Science Foundation, Grant No. DMR-1719875).

## Author contributions

J.L. and Z.G. designed and fabricated the devices. J.L. and Z.G. performed the device characterization. S.X and Q.H. assisted in the device fabrication. S.X., M.L., K.Z., U.J., R.L., and J.S. assisted in experiments. J.L., Z.G., and Q.L. wrote the manuscript with contributions from all authors. Q.L. supervised the project. Q.L. conceived the concept.

## Competing interests

The authors declare no competing interests.
