## [Peer Review File · Nature Communications]

REVIEWER COMMENTS

Reviewer #1 (Remarks to the Author):

In the paper titled “Electrically empowered microcomb laser”, the authors demonstrated an external cavity laser that can generate combs due to the combination of Kerr nonlinearity and EO modulation. This concept is novel, and the experimental results are solid. I think this paper can be accepted by Nature Communications after addressing the following issue:

1. I think some claims of the paper can be modified. Personally, I don't think this work addresses all the problems of “complex soliton initialization, high threshold, low power efficiency, and limited comb reconfigurability” for microcombs. It might be better to focus more on the dynamics of this unique laser configuration.
2. When calculating the wall-plug efficiency, I think the RF power applied to the cavity needs to be counted. One major problem for people using EO combs is the large RF power required. I think discussions regarding this problem should be added.
3. Can the author discuss more about the power of individual comb lines? What really matters in most of the applications is the comb line power, so a more detailed discussion could be helpful.
4. I feel that the statement “100% utilization of optical power fully contributing to comb generation” is strange. How to define which portion of power contribute to comb generation? For traditional microcomb this can be defined by the soliton state power/pump power, but I don't think it applies here.
5. What is the time domain pattern of this mode locked state? Is it more like a soliton or dark pulse? What is the pulse width?
6. Can the author add discussions about the tuning range of the FSR?

Reviewer #2 (Remarks to the Author):

This paper introduces a novel method of microcomb laser by hybridizing Kerr nonlinearity, EO modulation and the gain/lasing process in an EO modulated LN microresonator. The proposed scheme is novel and elegant in terms of the underlying concept, and exhibits significant performance advantages including high efficiency and a straightforward triggering process. While I do have some questions regarding the theoretical analysis detailed below, I believe a properly revised version could be considered for publication in Nature Communications.

1. Although the combination of Kerr nonlinearity, EO modulation and laser is intriguing, the concept of microcomb laser has previously been demonstrated using Kerr nonlinearity and lasing process [18,21]. Compared with these previous works, this comb spectra in this work show bifurcated spectral shapes

and narrower optical bandwidths. It would be great if the authors could provide more explanations and insights, from physics point of view, on the difference between these two microcomb laser schemes.

2. This method eliminates the need for pre-configuration compared with self-injection locking for microcomb stimulation. However, the proposed method require precise alignment of multiple resonances within a wideband between the two cavities, which seems quite challenging. The authors also mentioned, “this problem can be resolved by further optimization of the roundtrip length of the main laser cavity and introducing tunability”. Compared with self-injection locking, which involves tuning the optical phase - an easy task in integrated photonics, this method requires tuning the cavity length or group index, which is actually highly challenging for on-chip photonics? Please comment on these problems and potential solutions, if any.

3. The authors identified three limitations in previous microcomb works, including complex initialization process, low power efficiency and limited reconfigurability. The proposed scheme does solve the first two problems quite obviously, but I’m not too sure what the authors refer to when mentioning "reconfigurability". Does this refer to the fast chirping depicted in Fig. 4, or the comb spacing tuning shown in Fig. 3? Since controlling comb spacing and spectral shape is well known technique in Kerr-based microcombs, I hope the authors could be more specific when mentioning "reconfigurability" and provide more explanations if possible.

4. The linewidth testing shown in Fig. 2e, which is obtained from a heterodyne measurement of the entire comb, is quite confusing to me. The authors claim that the tested linewidth is “the upper limit of the intrinsic linewidth of individual comb lines” (Page 4, line 222)? In my understanding, the noises of comb lines with higher signal powers would overwhelm those of low-power comb lines. As a result when normalizing the noise spectrum of the entire comb, one is probably seeing a larger noise contribution from higher-power lines, which are not necessarily better or worse than the lower-power lines. Please provide more references or proof to support this claim.

5. Following up on the above question, it would be really nice if the authors could characterize the linewidths of each comb lines. The intrinsic linewidth variation among different comb lines could assist the understanding of the microcomb dynamics. For Kerr microcombs and EO combs, they both show the increasing linewidths when moving from the center to sides (Ref. [1, 2] below). I am curious about the linewidth variation trend of the proposed microcomb under different states as shown in Fig. 2b and Fig. 5d. Specifically, for the ‘clustered’ comb state shown in Fig. 2b, Fig. S3b and d, would the linewidth increase from respective centers of different clusters?

[1] Lei, F., Ye, Z., Helgason, Ó. B., Fülöp, A., Girardi, M., & Torres-Company, V. (2022). Optical linewidth of soliton microcombs. *Nature Communications*, 13(1), 3161.

[2] Skehan, J. C., Naveau, C., Schroder, J., & Andrekson, P. (2021). Widely tunable, low linewidth, and high power laser source using an electro-optic comb and injection-locked slave laser array. *Optics Express*, 29(11), 17077-17086.

6. The optical spectra presented in the experiment and simulation exhibit notable differences. Could the authors comment on the main reasons for the difference? Would the mismatch between the main laser

cavity and the racetrack resonator influence the comb state? Will the comb power keep increasing with higher RF power as claimed in Page 4 line 234? As the authors emphasize that the cavities are dispersion-engineered, what is the influence of the dispersion of different cavities (the racetrack cavity and the main laser cavity) on the comb states? Comprehensive analyses are needed to get a better understanding of the proposed microcomb.

7. In Fig. 2, the hybrid microcomb produces clustered microcombs, which are also shown in Fig. S3, while the spectral shapes in Fig. 5 are quite different. Could the authors comment on the possible origins of this difference. Is this due to the use of an optoelectronic feedback loop?

8. In Fig. 5, the authors show the feedback mode-locking of the comb laser. The concept is similar to the combination of EO combs and optoelectronic oscillators as demonstrated in previous works, such as Ref. [3-4] below. It is recommended to reference these works.

[3] Peng, H., Lei, P., Xie, X., & Chen, Z. (2021). Dynamics and timing-jitter of regenerative RF feedback assisted resonant electro-optic frequency comb. *Optics Express*, 29(26), 42435-42456.

[4] Sakamoto, T., Kawanishi, T., & Izutsu, M. (2006). Optoelectronic oscillator using a LiNbO₃ phase modulator for self-oscillating frequency comb generation. *Optics letters*, 31(6), 811-813.

9. There are several typos in describing Fig. 4:

Page 5, line 290-291: "Fig. 4f-h show the temporal variation of the 15-GHz beating signal..." While the temporal variations are presented in Fig. 4b-d.

Page 5, line 307-309: "The frequency tuning range of 1.2 GHz at the modulation speed of 100 MHz (Fig. 4h) corresponds to a frequency tuning rate..." The fast-chirping results are given in Fig. 4d.

Page 5, line 314-315: "As shown in Fig. 4e, the device exhibits a frequency tuning efficiency..." The efficiency is shown in Fig. 4h.

Author's response to Manuscript NCOMMS-23-59501-T:

Dear Editors and Reviewers of *Nature Communications*,

We greatly appreciate your thorough editing of our manuscript and we thank the reviewers for their critical reading of our manuscript and making suggestions that have improved the manuscript. We have revised the manuscript wherever necessary. We list reviewers' comments in blue and our responses in black.

Response to Reviewer 1's comments:

In the paper titled "Electrically empowered micro comb laser", the authors demonstrated an external cavity laser that can generate combs due to the combination of Kerr nonlinearity and EO modulation. This concept is novel, and the experimental results are solid. I think this paper can be accepted by Nature Communications after addressing the following issue.

RE: We thank the reviewer for the positive comments.

1. I think some claims of the paper can be modified. Personally, I don't think this work addresses all the problems of "complex soliton initialization, high threshold, low power efficiency, and limited comb reconfigurability" for microcombs. It might be better to focus more on the dynamics of this unique laser configuration.

RE: We thank the reviewer for the comment. We have revised the related sentences in the Abstract to make it clear. The analysis of the dynamics of our system is provided in detail in the supplementary information.

2. When calculating the wall-plug efficiency, I think the RF power applied to the cavity needs to be counted. One major problem for people using EO combs is the large RF power required. I think discussions regarding this problem should be added.

RE: We thank the reviewer for the comment. We understand that, for a commercial laser product, the overall wall-plug efficiency should include all power consumptions of the whole laser system to quantify its overall energy efficiency. However, such a metric does not provide insight into the essential performance of the semiconductor laser itself since a laser system always requires certain but different peripheral control circuits for proper operation that do not contribute to the final laser output power. This will make it difficult for readers to tell the fundamental laser performance and compare one with another. As such, the term of "wall-plug efficiency" is generally used to characterize the fundamental power efficiency of the diode laser itself, defined as the laser optical power compared with the electric power used to drive the laser diode. One typical example is that nearly all semiconductor lasers require temperature control, but the related power consumption is not included in the calculation of "wall-plug efficiency" in the literature.

We follow this convention in our paper. But to make it clear, we have added a sentence in the section of Comb Laser Performance to describe how the wall-plug efficiency is measured. Moreover, the detailed RF power applied to the device is provided in Fig.S3 of the supplementary information.

3. Can the author discuss more about the power of individual comb lines? What really matters in most of the applications is the comb line power, so a more detailed discussion could be helpful.

RE: We thank the reviewer for raising this question. The power of individual comb lines is in the range of 0.25—2mW. We have added this information in the section of Comb Laser Performance.

4. I feel that the statement "100% utilization of optical power fully contributing to comb generation" is strange. How to define which portion of power contributes to comb generation? For traditional microcomb this can be defined by the soliton state power/pump power, but I don't think it applies here.

RE: We thank the reviewer for the comment. We used this sentence to show the fact that, in our laser, all power of the optical wave stays in the form of comb output, in contrast to traditional microcomb in which only a small portion of the optical power is transferred to the comb from the pump laser. To make it clear, we have added a sentence to explain this in the Introduction section.

5. What is the time domain pattern of this mode-locked state? Is it more like a soliton or dark pulse? What is the pulse width?

RE: It is soliton-like pulses, as evident by the auto-correlation traces shown in the insets of Fig. 2b and Fig. 3b&c. The pulse width is estimated from the autocorrelation trace to be around 7 ps.

6. Can the author add discussions about the tuning range of the FSR?

RE: We thank the reviewer for the suggestion. The FSR of the comb can be tuned by about 1GHz. The details are provided in Fig.S3 of the supplementary information and are discussed in Section II.B.

Response to Reviewer 2's comments:

This paper introduces a novel method of microcomb laser by hybridizing Kerr nonlinearity, EO modulation and the gain/lasing process in an EO modulated LN microresonator. The proposed scheme is novel and elegant in terms of the underlying concept, and exhibits significant performance advantages including high efficiency and a straightforward triggering process. While I do have some questions regarding the theoretical analysis detailed below, I believe a properly revised version could be considered for publication in Nature Communications. This paper introduces a novel method of microcomb laser by hybridizing Kerr nonlinearity, EO modulation and the gain/lasing process in an EO modulated LN microresonator. The proposed scheme is novel and elegant in terms of the underlying concept, and exhibits significant performance advantages including high efficiency and a straightforward triggering process. While I do have some questions regarding the theoretical analysis detailed below, I believe a properly revised version could be considered for publication in Nature Communications.

RE: We thank the reviewer for the positive comments.

1. Although the combination of Kerr nonlinearity, EO modulation and laser is intriguing, the concept of microcomb laser has previously been demonstrated using Kerr nonlinearity and lasing process [18,21]. Compared with these previous works, this comb spectra in this work show bifurcated spectral shapes and narrower optical bandwidths. It would be great if the authors could provide more explanations and insights, from physics point of view, on the difference between these two microcomb laser schemes.

RE: We thank the reviewer for the comment.

The fundamental mechanism of comb generation in our laser is very different from that in Ref.[18,21] where the self-emergence of the soliton-state relies crucially on the thermal-optic nonlinearity of the erbium-doped fiber amplifier (EDFA) to compensate for that in the nested microresonator. In contrast, our laser uses EO modulation to initiate the comb generation, Kerr nonlinearity to broaden the comb spectrum and phase lock comb lines, and the laser gain to sustain the comb operation. We have explained this point clearly in the sections of Introduction and Comb laser performance. To make it clearer, we have added a sentence in the section of Comb laser performance.

The narrower comb spectral bandwidth in our laser is simply due to the lower pump power available in our gain chip compared with an EDFA. We have provided a discussion about comb spectral bandwidth in the Discussion section. As to the bifurcated comb spectrum, its exact physical nature is not clear at this moment, which requires future exploration. We speculate that it could be likely related to the dispersion and mismatched group delay. Note that our laser can also produce single-lobe comb spectrum. The details are shown in Fig.S3 of the supplementary information, as well as Fig.5d of the main text.

2. This method eliminates the need for pre-configuration compared with self-injection locking for microcomb stimulation. However, the proposed method require precise alignment of multiple resonances within a wideband between the two cavities, which seems quite challenging. The authors also mentioned, “this problem can be resolved by further optimization of the roundtrip length of the main laser cavity and introducing tunability”. Compared with self-injection locking, which involves tuning the optical phase - an easy task in integrated photonics, this method requires tuning the cavity length or group index, which is actually highly challenging for on-chip photonics? Please comment on these problems and potential solutions, if any.

RE: We thank the reviewer for raising this point. Controlling the cavity length can be realized by multiple approaches. Two examples are given below:

- I) Heterogenous integration approach where the gain element is bonded on the top of the external laser cavity. In this approach, the cavity length is purely determined by the external laser cavity, which can be precisely controlled by the fabrication process.
- II) Tuning approach to add a group-delay tuning element into the external laser cavity. One example is Ref.[1] given below.

[1] Y. Liu, *et al*, “Continuously tunable silicon optical true-time delay lines with a large delay tuning range and a low delay fluctuation,” *Opt. Express* 32, 7848 (2024).

We have added these points in the section of Discussion.

3. The authors identified three limitations in previous microcomb works, including complex initialization process, low power efficiency and limited reconfigurability. The proposed scheme does solve the first two problems quite obviously, but I’m not too sure what the authors refer to when mentioning "reconfigurability". Does this refer to the fast chirping depicted in Fig. 4, or the comb spacing tuning shown in Fig. 3? Since controlling comb spacing and spectral shape is well known technique in Kerr-based microcombs, I hope the authors could be more specific when mentioning "reconfigurability" and provide more explanations if possible.

RE: We thank the reviewer for the comment. The reconfigurability refers both the fast chirping and comb spacing switching and tuning. For fast chirping, we have demonstrated unprecedented chirping rate up to 2.4×10^{17} Hz/s orders of magnitude faster than other approaches. For comb spacing switching and tuning, we achieved it with high-speed EO modulation without involving complex tuning dynamics, which, again, is significantly faster than conventional Kerr microcombs. To make it clear, we have revised the related sentences in Abstract and Introduction.

4. The linewidth testing shown in Fig. 2e, which is obtained from a heterodyne measurement of the entire comb, is quite confusing to me. The authors claim that the tested linewidth is “the upper limit of the intrinsic linewidth of individual comb lines” (Page 4, line 222)? In my understanding, the noises of comb lines with higher signal powers would overwhelm those of low-power comb lines. As a result when normalizing the noise spectrum of the entire comb, one is probably seeing a larger noise contribution from higher-power lines, which are not necessarily better or worse than the lower-power lines. Please provide more references or proof to support this claim.

RE: We thank the reviewer for raising this point. The reviewer is correct and we have revised the relevant sentences in the paper. We provide more detailed linewidth characterizations in the following.

5. Following up on the above question, it would be really nice if the authors could characterize the linewidths of each comb line. The intrinsic linewidth variation among different comb lines could assist the understanding of the microcomb dynamics. For Kerr microcombs and EO combs, they both show the increasing linewidths when moving from the center to sides (Ref. [1, 2] below). I am curious about the linewidth variation trend of the proposed microcomb under different states as shown in Fig. 2b and

Fig. 5d. Specifically, for the ‘clustered’ comb state shown in Fig. 2b, Fig. S3b and d, would the linewidth increase from respective centers of different clusters?

[1] Lei, F., Ye, Z., Helgason, Ó. B., Fülöp, A., Girardi, M., & Torres-Company, V. (2022). Optical linewidth of soliton microcombs. *Nature Communications*, 13(1), 3161.

[2] Skehan, J. C., Naveau, C., Schroder, J., & Andrekson, P. (2021). Widely tunable, low linewidth, and high power laser source using an electro-optic comb and injection-locked slave laser array. *Optics Express*, 29(11), 17077-17086.

RE: We thank the reviewer for the suggestion. We have performed detailed characterizations on the linewidths of individual comb lines. We used the setup shown in Fig. R1 below for the measurements, in which an individual comb line is separated from the rest of the comb by a tunable narrow-band fiber Bragg grating (FBG) filter and its linewidth is measured by the self-heterodyning method.

Figure R1 Schematic of the experimental setup used to characterize the linewidths of individual comb lines.

Figure R2a shows the overall frequency noise spectra for different lasing states. For the comb states, the linewidth measurement was performed on the entire combs similar to Fig. 2 in the paper. It shows an overall linewidth of about 7 kHz, 7 kHz, and 1.5 kHz, respectively, for the single-mode lasing state, the single-lobe comb state, and the bifurcated two-lobe comb state. These values are slightly worse than those shown in the paper simply due to the slight degradation of the LN device and the RSOA in the past four months.

The recorded linewidths for individual comb lines are shown in Fig. R2c and d for the two comb states. Due to the limited powers of the individual comb lines, we can only perform linewidth characterizations for the central portion of the combs. Figure R2c shows that, for the single-lobe comb state, the individual comb lines exhibit linewidths in the range of 1–3 kHz. Figure R2d shows that one lobe of the comb exhibits individual comb linewidths in the range of 0.8–2kHz, while the other lobe exhibits slightly larger linewidths in the range of 5–8kHz. The exact physical reason for this difference is not clear at this moment, which requires further exploration.

Overall, the linewidths of individual comb lines do not show a certain deterministic trend across the comb spectrum, distinctive to the ones shown in Ref. [1,2] pointed out by the reviewer. This is understandable since the linewidth feature shown in Ref.[1] is due to the recoil effect between Raman-induced self-frequency shift and dispersive wave generation, and that shown in Ref.[2] is due to the noise multiplication during the cascaded sideband creation in a pure EO comb. The mechanism of comb generation is fundamentally different in our comb laser which could lead to different linewidth behaviors.

Figure R2 a) Frequency noise spectrum of a single-mode lasing state, single-lobe comb state (Comb 1), and bifurcated two-lobe comb state (Comb 2). For the combs, the frequency noise is for the entire combs, similar to Fig.2 in the paper. b) Optical spectrum (blue) and recorded linewidth (red dot) for the single-mode lasing state. c) Optical spectrum (blue) and recorded linewidths (red dots) of individual comb lines, for the single-lobe comb state (Comb 1). d) Same as c) but for the bifurcated two-lobe comb state (Comb 2). In b)-d), the small spectral gap on the noise floor is due to the spectral filtering by the FBG filter.

6. The optical spectra presented in the experiment and simulation exhibit notable differences. Could the authors comment on the main reasons for the difference? Would the mismatch between the main laser cavity and the racetrack resonator influence the comb state? Will the comb power keep increasing with higher RF power as claimed in Page 4 line 234? As the authors emphasize that the cavities are dispersion-engineered, what is the influence of the dispersion of different cavities (the racetrack cavity and the main laser cavity) on the comb states? Comprehensive analyses are needed to get a better understanding of the proposed microcomb.

RE: We thank the reviewer for the detailed questions. We answer the questions separately in the following:

I) Could the authors comment on the main reasons for the difference?

In fact, we are able to generate a comb from Laser β similar to the simulation, as we showed in Fig.S2 of the supplementary information. The mechanism underlying the bifurcated comb produced by Laser α is currently still under investigation, which may result from complex physics not included in the simplified model. We have added one sentence in Section III.B of the supplementary information to make this point clear.

II) Would the mismatch between the main laser cavity and the racetrack resonator influence the comb state?

Yes, the mismatch has strong impact on the stability and shape of the comb state. Chaotic comb may occur in a mismatched laser. We are currently still investigating the dynamics of the laser at different cavity length offset.

III) Will the comb power keep increasing with higher RF power as claimed in Page 4 line 234?

The detailed relationship between the produced comb laser power and the RF driving power is shown in Fig. S3 a,c,f,h of the supplementary information for the two lasers. We observed comb power saturation in Laser β . In Laser α , however, we haven't observed saturation within the applicable RF power range.

IV) As the authors emphasize that the cavities are dispersion-engineered, what is the influence of the dispersion of different cavities (the racetrack cavity and the main laser cavity) on the comb states?

Dispersion of the racetrack microresonator is crucial for the comb generation and mode locking. A slight anomalous dispersion is required, as we showed in the numerical modeling. The dispersion of the main laser cavity plays a minor role. We engineered it to be close to zero.

7. In Fig.2, the hybrid microcomb produces clustered microcombs, which are also shown in Fig. S3, while the spectral shapes in Fig. 5 are quite different. Could the authors comment on the possible origins of this difference. Is this due to the use of an optoelectronic feedback loop?

RE: We thank the reviewer for the question. We do observe different comb spectra for RF driven comb and feedback lock comb. Currently, the feedback locked comb is wider in spectrum. This could be related to that the dynamically adjusted feedbacked RF modulation assist in the generation of broader comb, but the detailed mechanism require further investigation in future works.

8. In Fig. 5, the authors show the feedback mode-locking of the comb laser. The concept is similar to the combination of EO combs and optoelectronic oscillators as demonstrated in previous works, such as Ref. [3-4] below. It is recommended to reference these works.

[3] Peng, H., Lei, P., Xie, X., & Chen, Z. (2021). Dynamics and timing-jitter of regenerative RF feedback assisted resonant electro-optic frequency comb. *Optics Express*, 29(26), 42435-42456.

[4] Sakamoto, T., Kawanishi, T., & Izutsu, M. (2006). Optoelectronic oscillator using a LiNbO₃ phase modulator for self-oscillating frequency comb generation. *Optics letters*, 31(6), 811-813.

RE: We thank the reviewer for the suggestion. We have added these two references as Ref.[47, 48] in the paper.

9. There are several typos in describing Fig. 4: Page 5, line 290-291: "Fig. 4f-h show the temporal variation of the 15-GHz beating signal..." While the temporal variations are presented in Fig. 4b-d. Page 5, line 307-309: "The frequency tuning range of 1.2 GHz at the modulation speed of 100 MHz (Fig. 4h) corresponds to a frequency tuning rate..." The fast-chirping results are given in Fig. 4d. Page 5, line 314-315: "As shown in Fig. 4e, the device exhibits a frequency tuning efficiency..." The efficiency is shown in Fig. 4h.

RE: We thank the reviewer for finding these typos. We have corrected them in the paper.

REVIEWERS' COMMENTS

Reviewer #1 (Remarks to the Author):

The authors have addressed all my comments and I suggest publication for this paper.

Reviewer #2 (Remarks to the Author):

The authors have addressed most of my comments. There is still a lot of physics here unexplored and not yet fully understood but I think it is reasonable to leave them to future studies as the results here are quite interesting and deserve to be published. There is one typo though:

The unit of the phase noise in Fig. 5g should be "dBc/Hz".